# Polymer-Based Wound Dressings Loaded with Ginsenoside Rg3

**DOI:** 10.3390/molecules28135066

**Published:** 2023-06-28

**Authors:** Jiali Yang, Lifeng Zhang, Xiaojuan Peng, Shuai Zhang, Shuwen Sun, Qiteng Ding, Chuanbo Ding, Wencong Liu

**Affiliations:** 1College of Chinese Medicinal Materials, Jilin Agricultural University, Changchun 130118, China; yjl1104481279@163.com (J.Y.); zlff0429@163.com (L.Z.); pxj15083336533@163.com (X.P.); zhangshuai4389@163.com (S.Z.); ssw170331@163.com (S.S.); ding152778@163.com (Q.D.); 2College of Traditional Chinese Medicine, Jilin Agriculture Science and Technology College, Jilin 132101, China; 3School of Food and Pharmaceutical Engineering, Wuzhou University, Wuzhou 543003, China

**Keywords:** wound dressing, ginsenoside Rg3, hydrogel, nanofiber

## Abstract

The skin, the largest organ in the human body, mainly plays a protective role. Once damaged, it can lead to acute or chronic wounds. Wound healing involves a series of complex physiological processes that require ideal wound dressings to promote it. The current wound dressings have characteristics such as high porosity and moderate water vapor permeability, but they are limited in antibacterial properties and cannot protect wounds from microbial infections, which can delay wound healing. In addition, several dressings contain antibiotics, which may have bad impacts on patients. Natural active substances have good biocompatibility; for example, ginsenoside Rg3 has anti-inflammatory, antibacterial, antioxidant, and other biological activities, which can effectively promote wound healing. Some researchers have developed various polymer wound dressings loaded with ginsenoside Rg3 that have good biocompatibility and can effectively promote wound healing and reduce scar formation. This article will focus on the application and mechanism of ginsenoside Rg3-loaded dressings in wounds.

## 1. Introduction

The skin, the largest organ of the human body [1,2], is mainly divided into three parts: epidermis, dermis, and subcutaneous tissue, which can effectively protect the internal environment and prevent microbial invasion [3,4,5,6,7,8]. Once damaged, there will be many local or systemic problems, and the body will begin the process of wound healing. If acute injury is not properly treated, it can form a chronic, non-healing wound or proliferative scar tissue [9,10,11].

Wound healing involves a series of complex biological processes [12]. At this point, good wound management measures are crucial for wound healing. Currently, wound management measures include debridement, infection control, and wound dressings [12,13,14,15]. An ideal wound dressing should not only maintain the moist environment required for wound healing [16,17,18], but also have good breathability [19,20]. In addition, it should also have good antibacterial ability and biocompatibility [21,22,23,24]. The common drawbacks of wound dressings include poor antibacterial activity, weak antioxidant capacity, poor biodegradability, poor mechanical properties, and the need for frequent replacement [25].

Wound dressings can be composed of biopolymers and synthetic polymers to achieve complementary effects. Some biopolymers used for dressings contain cellulose, chitosan, gelatin, etc. [26,27,28], which have good biocompatibility and biodegradability. The artificial polymers used for dressings mainly contain polyvinyl alcohol (PVA), ε-Caprolactone (PCL), polylactic acid (PLA), polyvinylpyrrolidone (PVP), and other compounds [29,30,31,32], which have excellent mechanical properties.

The polymer wound dressing mentioned above can enhance its effectiveness by loading natural bioactive substances. Ginsenoside Rg3 (Rg3) is a steroidal glycoside obtained from ginseng roots [33], which has rich pharmacological activities, such as anti-inflammatory, antibacterial, antioxidant, anti-tumor, and hypoglycemic [34,35,36,37,38,39]. Due to its biocompatibility, it has attracted widespread attention from many researchers. However, its poor water solubility limits its application in biomedicine. Therefore, it can be loaded into various polymer materials to promote its application. This article reports the applications and mechanism of Rg3 polymer-based dressings in wounds.

## 2. Classification of Wounds

Wounds are skin structural damage caused by injury [40,41] and could be classified into acute and chronic wounds according to their duration and the nature of the healing process [42]. Acute wounds can recover in a short period of time because the acute wound micro-environment builds a well-organized ECM and a substrate synthesis rate that exceeds its degradation. However, the imbalance in this phase will result in a chronic, non-healing wound [43,44,45]. Some examples of chronic wounds include diabetes wounds, burn wounds, ulcer wounds, etc. [46,47], which exhibited the failure to heal in time through the normal wound-healing process [48,49,50].

## 3. Wound-Healing Process

Wound healing involves complex physiological processes that require the participation of many different types of cells and various growth factors [51,52,53]. The process of wound healing is mainly divided into the following four stages: hemostasis, inflammation, proliferation, and remodeling [54,55,56]. These stages are not independent and coordinate to restore tissue integrity and homeostasis in a harmonious manner [57,58,59,60].

### 3.1. Hemostasis Phase

The hemostasis stage occurs immediately after injury and is mainly divided into three stages: vasoconstriction, primary hemostasis, and secondary hemostasis, which are achieved by mechanical means [61,62,63].

### 3.2. Inflammation Phase

The inflammatory stage occurs immediately after injury, usually lasting for 3 days. At this stage, basophils and mast cells excrete histamine, serotonin, and other products, leading to vasodilation and increased vascular permeability, which develops into inflammation [64,65]. Dendritic cells (DCs) are involved in antigen presentation that triggers T-cell responses [66]. B lymphocytes secrete antibodies, produce various cytokines and growth factors, antigen presentation, regulate T-cell activation and differentiation, and regulate lymphoid tissue to affect the immune response [67]. Platelet-derived growth factor (PDGF), transforming growth factor-β (TGF-β) with fibroblast growth factor (FGF), and neutrophils begin to aggregate at the site of injury [68]. Neutrophils act as the first line of defense to prevent microbial invasion. After 3–4 days, macrophages clear depleted neutrophils to avoid non-specific tissue damage and sustained inflammation [54].

### 3.3. Proliferation Phase

On the 4th day of wound healing, the proliferation stage begins, which lasts for a long time and is mainly divided into tissue granulation formation, angiogenesis, and wound contraction [69]. Collagen and fibronectin deposit to form granulation tissue [70]. Macrophages and endothelial cells initiate angiogenesis in response to pro-angiogenic signals and activate tissue growth [71,72,73]. In granulation tissue, keratinocytes migrate, proliferate, differentiate, and start the re-epithelialization process [74,75]. Fibroblast differentiation into myofibroblasts promotes wound healing [47].

### 3.4. Remodeling Phase

After three weeks of wound formation, the remodeling stage begins, which is the final stage of wound healing, also regarded as the mature stage. During this stage, different types of wounds have different durations [76,77]. The wound is covered by fibroblasts, which promote collagen rearrangement and enhance the tensile strength of the tissue [78]. Finally, a fully mature scar is formed [79,80].

## 4. Wound Dressings

The wound dressings used for wound management are mainly divided into traditional dressings, modern dressings, and bioactive dressings [81,82]. Traditional dressings are the most commonly used dressings because of their low cost and simple preparation process [60]. They are responsible for isolating the wound from the outside world, and they mainly include gauze, bandages, and plaster [83]. However, some drawbacks limit their application, such as difficulty in maintaining a humid environment, poor biodegradability, and the dressing needing to be frequently replaced, which can cause inconvenience to patients and may even cause further skin damage [84,85,86,87,88].

In contrast, a wide range of modern dressings not only maintain a moist environment for wounds, but also have good biocompatibility, biodegradability, and water vapor permeability [89,90,91]. These dressings are usually prepared from natural or synthetic polymers, such as hydrogels, nanofibers, films, and foam. Yet, their biological activity is limited.

Bioactive dressings are used to deliver bioactive molecules, such as antibiotics and growth factors [92]. Antibiotics can prevent wound infections, and growth factors can restore vitality to damaged tissues. Examples of biologically active dressings include foam, hydrogel, and nanofiber, which are usually prepared with natural or artificial polymers, such as gelatin, PVA, and PVP [25]. Rg3 can be loaded into these dressings because it has some effective pharmacological properties, such as anti-inflammation and promoting wound healing.

## 5. Biological Activity of Rg3

Ginseng, the root of *Panax ginseng* C.A. Meyer, has been widely used in East Asia countries for thousands of years as a natural supplement [93]. Pharmacognostic research on ginseng, the root of *Panax ginseng* Meyer, has shown that ginsenosides, triterpenoid saponins, are the bioactive components of ginseng [94,95]. More than 100 types of ginsenosides have been isolated and determined from ginseng [96]. Ginsenoside is one of the main secondary metabolites in ginseng and has various pharmacological effects [97,98,99,100]. Ginsenoside Rg3 accounts for 4.7% of all saponins, which have been proven to have multiple pharmacological effects, such as anti-inflammatory, analgesic, anti-tumor, anti-diabetes, anti-obesity, anti-depression, anti-melanogenesis, liver protection, cardiovascular protection, immune promotion, anti-fatigue, etc. [101,102,103,104,105,106,107,108,109,110,111,112].

Rg3 can promote wound healing, possibly due to its anti-inflammatory [101], antibacterial [102], and antioxidant activities [103]. Oxidative stress is one of the factors that delay wound healing, and some antioxidant therapies promote chronic wound healing by loading antioxidants into dressings to eliminate reactive oxygen species (ROS). NRF2 can activate antioxidant responses and is a major participant in the antioxidant and anti-inflammatory signaling pathways [113,114]. NRF2 has also been shown to increase the expression of heme oxygenase (HO-1) [115]. A recent study suggests that Rg3 can inhibit oxidative stress damage to human skin fibroblasts by reducing the production of ROS, increasing the level of NRF2, and promoting the expression of HO-1. In addition, it is reported that Rg3 can play a protective role in its antioxidant capacity by increasing the level of cellular antioxidant enzymes and acting as a free radical scavenger [116,117]. The antioxidant capacity of Rg3 may be related to its steroid structure, as it has been reported that steroid drugs have good antioxidant activity.

Rg3 also interacts with many molecular targets involved in inflammatory responses. A recent study has shown that Rg3 can reduce the production of histamine and pro-inflammatory cytokines, such as IL-1β and TNF-α, and alleviate allergic reactions in mice by inhibiting the MAPK/NF-κB signaling pathway. The mitogen-activated protein kinase (MAPK) signaling pathway regulates the whole cellular process, containing the activation and differentiation of immune cells [118]. In allergic responses, MAPK signaling is related to nuclear factor kappa B (NF-κB) activation, and NF-κB can control the transcription of pro-inflammatory cytokines [119]. The anti-inflammatory effect of Rg3 can be attributed to its steroid structure, and it has been reported that steroid drugs have excellent anti-inflammatory activity [120,121], and they have been widely used in inflammatory diseases because they can inhibit the expression of pro-inflammatory cytokines IL-1, IL-6, and TNF-α.

## 6. Application of Ginsenoside Rg3 in Wound-Healing Course

In previous animal experiments, we found that Rg3 can restore the structure and function of damaged skin and promote fibroblast proliferation and angiogenesis [122,123], while hypertrophic scars (HSs) are the development result of burns, wound infections, and various skin injuries [124,125,126,127]. Abnormal tissue and cellular structures and increased inflammation and angiogenesis are the main characteristics of HS formation [128,129,130,131]. Several studies [132,133,134,135] have confirmed that Rg3 is a promising new method for treating HS, which can significantly improve the thickness of the dermis and epidermis and promote scar-free wound healing, and its main mechanism is regulating the TGF-β/smads signaling pathway and the expression of angiogenic factors, such as VEGF and CD31, during the remodeling stage of the wound-healing process.

## 7. Rg3-Loaded Wound Dressing

The structure of Rg3 leads to its hydrophobicity, and its solubility is extremely low under physiological conditions [136,137]. Therefore, finding a suitable drug delivery system is very important. The application of Rg3-loaded dressings in promoting wound healing and inhibiting scar formation are now shown in Table 1.

### 7.1. Hydrogels

Hydrogels are hydrophilic three-dimensional polymer wound dressings. They can absorb a large amount of wound exudates to keep the wound moist [144] and have good biocompatibility and biodegradability. They have been widely used to load various therapeutic agents to promote wound healing [144]. However, they also have some limitations, such as poor mechanical properties in the swelling state, which can be improved by a combination of natural and synthetic polymers. Several researchers have reported polymer-based hydrogels loaded with Rg3 as showed in Figure 1.

In order to improve the bioavailability of Rg3, Longhai Jin et al. [122] used Methoxy poly(ethylene glycol) (PEG), D,L-lactide (D,L-LA), and glycolide (GA) as a matrix to load Rg3 to treat perianal ulcers in rats. The encapsulation efficiency of Rg3 was 10%. The experimental results showed that the prepared hydrogel had a uniform porous structure; Rg3 could be released slowly for 120 h continuously; and the biocompatibility of the hydrogel was good, without obvious cytotoxicity to mouse skin fibroblasts. In addition, Rg3-loaded hydrogel has good biodegradability, can inhibit local and systemic inflammatory reactions, activates the ErK signaling pathway, and promotes the healing of perianal ulcers in rats.

Tao Zheng et al. [138] prepared Rg3-loaded sandwich hydrogel dressings with hydrophilic matrices, such as xanthan gum, hyaluronic acid sodium (HA), and carboxymethyl chitosan (CMCS). The loading amount of Rg3 is 220 μg/mL. With the change of hydrogel thickness, the release time of Rg3 can be adjusted to meet the needs of various wounds. The research results of rabbit ear infected wounds showed that Rg3-loaded sandwich hydrogel could reduce the inflammatory reaction caused by infection, promote the formation of granulation tissue, and promote scar-free wound healing.

In addition, Xiaojuan Peng et al. [123] prepared a thermosensitive hydrogel loaded with Rg3 with hydrophilic matrices, such as poloxamer 407(P407), HA, chitosan (CS), etc., wherein the loading amount of Rg3 is about 3 mg/mL, the hydrogel can continuously release Rg3, and the results show that Rg3 can reach a 40% release rate in 120 h. They also tested the antibacterial and antioxidant capacity of the temperature-sensitive hydrogel loaded with Rg3, which proved that the hydrogel loaded with Rg3 has good antibacterial and antioxidant activity and can significantly promote wound healing. Meanwhile, in vivo research results showed that Rg3 can enhance the expression of autophagy proteins by regulating the MAPK and NF-κB signaling pathways, and promote wound repair by regulating the diversity of wound microbiota.

The above studies all used different hydrophilic matrices to load Rg3 to improve its bioavailability, and the encapsulation efficiency reached a maximum of 10%. The results of the above studies in vivo show that Rg3-loaded hydrogel has a good promoting effect on wound healing, and its effect is mainly achieved by regulating multiple signaling pathways, such as the MAPK, NF-κB, and ErK signaling pathway, which can effectively reduce the expression of IL-1, IL-6, TNF-α, and other inflammatory factors through these signal pathways. Therefore, Rg3-loaded hydrogel can be widely used in promoting wound healing.

### 7.2. Nanofibers

Nanofiber dressing has high porosity and good water vapor permeability, which can provide a favorable environment for cellular respiration and gas permeation, and supports drug delivery, thus inhibiting microbial infection [145]. In addition, it has a high surface area and volume ratio, excellent mechanical properties, and can simulate natural extracellular matrix (ECM) [146,147,148]. Electrospinning is currently one of the most commonly used and effective technologies for preparing nanofiber. There are various characteristics in electrospinning technology containing its mild process, allowing drugs, proteins, and even DNA to be uniformly embedded in fibers without causing drug denaturation or deactivation [149,150]. Moreover, it can regulate drug release behavior by controlling the composition of nanofibers [151]. However, according to the composition of the polymer, the exudate absorption and other properties of the nanofibers also vary, and suitable polymers need to be used according to different clinical environments. Some researchers have reported on polymer nanofiber wound dressings loaded with Rg3 as showed in the Figure 2.

In order to control the long-term sustained release of Rg3 at the wound site and improve the utilization rate of Rg3, Liying Cheng et al. [133] used PLA to load Rg3 through electrospinning, which changed the crystalline state of Rg3, reaching an encapsulation rate of 10%, and Rg3 slowly released in the scaffold for 3 months. In addition, the research results indicate that the PLA fiber scaffold loaded with Rg3 is a transplantable controlled-release drug system that can control the drug release concentration in rabbits by changing the drug content added to the pre-spinning solution. From the potential mechanism of Rg3-loaded PLA scaffolds on wound healing, it can be seen that Rg3-loaded PLA scaffolds can significantly inhibit excessive inflammation, inhibit VEGF expression, and promote scar-free wound healing.

Inspired by mussels, Liying Cheng et al. [139] used electrospinning technology to prepare a polydopamine-coated PLGA cellulose scaffold, which was loaded with basic fibroblast growth factor (bFGF) to immobilize the bioactive drug Rg3. The encapsulation efficiency of Rg3 was about 6%, and it was slowly released in 40 days. This method is conducive to the biological functionalization of drug-loaded scaffolds, and is a good combination of drug therapy and biological signal stimulation. In vivo research results show that the scaffold not only affects the early stage of wound healing, but also plays a positive role in the later stage of wound healing.

In order to improve the hydrophobicity of electrospinning nanofibers, Xiaoming Sun et al. [140] used chitosan to coat nanofibers and prepared a hydrophilic PLGA electrospinning nanofiber membrane loaded with Rg3. The encapsulation efficiency of Rg3 reached 6%. Coating chitosan on the surface of electrospinning fiber can significantly improve the hydrophilicity of the fiber membrane, maintain the release rate of the Rg3 of the fiber membrane, and make it release slowly within 40 days. In addition, in vivo research results indicate that fiber scaffolds loaded with Rg3 can significantly promote scar-free wound healing. However, Xiaoming Sun et al. [132] used hyaluronic acid for coating and designed a hyaluronic acid-coated PLGA electrospinning fiber scaffold loaded with Rg3. The encapsulation efficiency of Rg3 also reached 6%, and the slow release of Rg3 within 40 days enhanced the biological activity of the fiber scaffold. In this study, the prepared hydrophilic fiber scaffold loaded with Rg3 has a stable fiber structure and long-lasting mechanical properties, which not only effectively promotes wound healing in the early stage, but also inhibits scar formation in the later stage, making it close to healthy skin.

Inspired by the physical and chemical environment of ECM, Tingting Xu et al. [141] developed a new photo-crosslinked nanofiber scaffold loaded with Rg3 using hydrophilic matrices, such as PEO, to promote wound healing. The encapsulation efficiency of Rg3 was 1%, and it can be slowly released within 16 days. The research results show that biomimetic fiber scaffolds modified with adhesive peptides can accelerate the proliferation of fibroblasts and effectively promote wound healing. In addition, it can inhibit the formation of HS by continuously releasing Rg3 in the later stage, reducing angiogenesis and collagen accumulation.

Wenguo Cui et al. [142] designed a poly(L-lactide) PLLA electrospun fiber scaffold loaded with Rg3 to cover the entire thickness of the skin resection site to restore the skin structure and function and inhibit the formation of HS. In this study, Rg3 was loaded onto the fiber scaffold in an amorphous state and easily dissolved from the scaffold, achieving a 10% encapsulation rate and sustained release for more than 30 days. In vivo experimental results showed that the scaffold can reduce the thickness of the dermis and the number of collagen fibers to inhibit the formation of HS.

Now, there are also wound dressings loaded with Rg3 applied to chronic wounds. XiongGuo et al. [143] used hydrophilic matrices, such as polyurethane and chitosan quaternary ammonium cation, as raw materials for electrospinning to prepare wound dressings loaded with Rg3. The encapsulation efficiency of Rg3 reached 2–4%, which can be slowly released within 35 days. In addition, the research results show that nanofibers loaded with Rg3 can promote the proliferation and migration of keratinocytes and fibroblasts and promote burn wound healing through potential mechanisms such as reducing inflammation and promoting vascular maturation. In particular, during the remodeling stage of wound healing, nanofibers loaded with Rg3 can prevent scar formation by regulating collagen deposition. This study confirms that nanofibers loaded with Rg3 can not only effectively promote burn wound repair, but also improve the aesthetics of wound healing, restoring injured skin to a state similar to that of healthy skin.

The above studies have all adopted different means to solve the solubility problem caused by the hydrophobicity of Rg3. Electrospinning can transform Rg3 from crystalline to amorphous, greatly improving its solubility. In addition, the introduction of hydrophilic matrices, such as PEO, HA, and CS, in electrospinning has also improved the bioavailability of Rg3, and the encapsulation efficiency has reached 1–10%. From the potential mechanism of action of Rg3-loaded nanofibers on wound healing, it can be seen that Rg3-loaded nanofibers mainly play a positive role in the inflammatory and proliferative stages of wound healing by regulating the expression of inflammatory factors and vascular growth factors. In addition, they also significantly inhibit scar formation during the remodeling stage of wound healing by regulating the levels of collagen deposition factors and vascular growth factors. Therefore, nanofibers loaded with Rg3 can be widely used in the field of wound repair, promoting scar-free wound healing.

## 8. Conclusions

In this review, we summarized the research on polymer dressings loaded with Rg3 and their mechanisms for promoting wound healing. These polymer dressings not only provide a new means to improve the bioavailability of Rg3, that is, by loading the hydrophobic drug Rg3 onto a hydrophilic matrix to achieve long-term controlled release. In addition, after loading Rg3, the biological activity of the dressings is greatly improved, which can promote wound healing and inhibit scar formation through the antioxidant, anti-inflammatory potential mechanisms and other mechanisms, such as regulating collagen deposition and angiogenesis. This provides new opportunities and ideas for high-activity and low-utilization drugs similar to Rg3, and contributes to the design and development of dressings suitable for wound healing.

Objectively speaking, the animal models used in most of the above studies are mice, rats, and rabbits, which have different skin morphology and wound-healing processes compared to humans. Moreover, the reported studies are mostly in the preclinical stage and require clinical trials.

It should be mentioned that the application direction of wound dressings loaded with Rg3 mentioned above is relatively single, mainly focusing on acute wound healing. There are few reports on other wound healing, especially chronic wound healing. In addition, due to its excellent biological activity, it can play a positive role in promoting bone repair, anti-tumor, and other fields. Therefore, these wound dressings loaded with Rg3 can be applied in other new fields.

In addition, the types of Rg3-loaded wound dressings described above also need to be broadened. We found that there are few studies on Rg3-loaded wound dressings at present, mainly focusing on hydrogels and electrospinning. At present, nanoparticles are also widely used in the field of wound healing. The high specific surface area of nanoparticles enables them to interact with the cell surface and enter the internal environment of cells, so as to achieve large therapeutic effects. However, compared with hydrogels, nanofibers, and other dressings, their drug loading may be a big test. Therefore, combining nanoparticles with hydrogels or electrospinning to form composite carriers to load Rg3 may be a feasible solution.

In the future, it is important to investigate the correlation between the activities of Rg3-loaded dressings and wound healing through a series of complete in vivo mechanism validation methods. At present, the research on the mechanism of Rg3-loaded wound dressings in the wound-healing process is simple, with a relatively single healing stage and relatively simple validation methods. The wound-healing process is composed of multiple stages, complex and intertwined, and high-flux technology is widely used in various fields. However, there are few reports on the promoting effect of Rg3-loaded polymer dressings on wound healing. At present, it can be found that some wound dressing materials loaded with Rg3 are made perfectly, with rich chemical synthesis and superior performance effects. However, subsequent mechanism research is not systematic. Conversely, pharmacological researchers’ understanding of materials is generally limited, and material selection and synthesis are relatively inefficient, which seems to form a situation that one understand much materials but little pharmacological mechanisms, and vice versa, as materials synthesis research and pharmacological mechanisms research are relatively complex. Therefore, close collaboration between synthetic chemists and pharmacological researchers is needed to better design and develop polymer dressings suitable for Rg3 loading and to elucidate their potential mechanisms for wound healing at the molecular level.

## Figures and Tables

**Figure 1 molecules-28-05066-f001:**
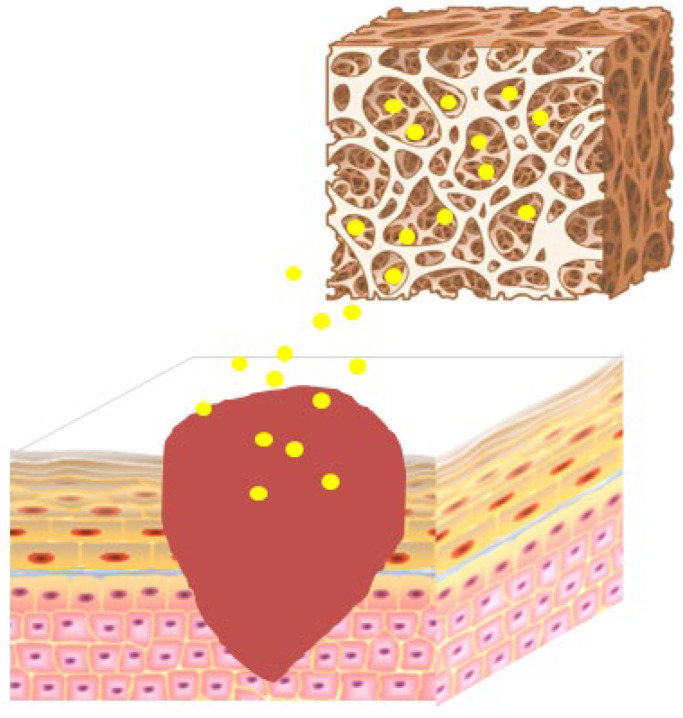
Application of Rg3-loaded hydrogel on skin wound, the yellow point represent Rg3.

**Figure 2 molecules-28-05066-f002:**
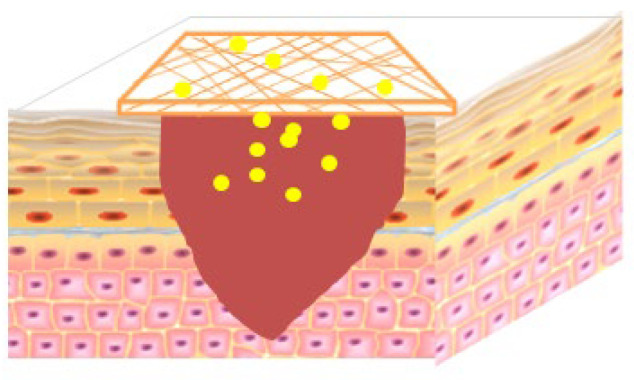
Application of Rg3-loaded fiber scaffold on skin wound, the yellow point represent Rg3.

**Table 1 molecules-28-05066-t001:** Application of Rg3-loaded dressings in promoting wound healing and inhibiting scar formation.

Serial Number	Dressing Type	Composition	Animal Model	Molecular Mechanism	Reference
1	hydrogel	Methoxy poly(ethylene glycol) (PEG); D,L-lactide (D,L-LA); glycolide (GA)	perianal ulcer in rats	Decrease the level of NF-κB, TNF-α, IL-1, and IL-6; Increase the expression of VEGF and CD31; Activate ERK signaling pathway;	[122]
2	hydrogel	Xanthan gum; hyaluronic acid sodium (HA); carboxymethyl chitosan(CMCS)	rabbit ear infected wounds	Increase the expression of α-SMA and CD31; Reduce the levels of pro-inflammatory factors	[138]
3	hydrogel	Poloxamer 407(P407); Hyaluronic Acid(HA); chitosan(CS)	back wound in mice	Increase the expression of HIF-1α, VEGF, CD31, and KRT; Inhibit the MAPK and NF-κB signal pathways; Increase the expression of LC3	[123]
4	fibrous membrane	Poly(L-lactide) (PLA)	rabbit ear wounds	Decrease the level of COL-1 and VEGF; Inhibit inflammation	[133]
5	fibrous membrane	Poly(D,L-lactide-co-glycolide) (PLGA); Poly (dopamine)	rabbit ear wounds	Decrease the level of COL-1, CD31, and VEGF	[139]
6	fibrous membrane	Poly(D,L-lactide-co-glycolide) (PLGA); Chitosan (CS)	rabbit ear wounds	Decrease the expression of COL-1 and VEGF	[140]
7	fibrous membrane	Poly(γ-glutamic acid) (γ-PGA); Polyethylene oxide(PEO)	back wound in rats	Decrease the level of CD31 and VEGF	[141]
8	fibrous membrane	Poly(D,L-lactide-co-glycolide) (PLGA); Hyaluronic Acid(HA)	rabbit ear wounds	Decrease the expression of COL-1	[132]
9	fibrous membrane	Poly(L-lactide) (PLLA)	rabbit ear wounds	Improve the collagen fiber density and micro vessels formation	[142]
10	fibrous membrane	polyurethane (PU); hydroxypropyl trimethyl ammonium chloride chitosan	third-degree burn wound in rats	Regulate the ratios of type I and III collagens; Increase the expression of α-SMA and CD31	[143]

## Data Availability

Data supporting the findings are available from the corresponding authors upon reasonable request.

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
