# Peer review of "Polymer-Based Wound Dressings Loaded with Ginsenoside Rg3"

_molecules, 2023, doi:10.3390/molecules28135066_

Round 1

Reviewer 1 Report

The authors have compiled recent works related to hydrogels and electrospun membranes containing ginsenoside Rg3. The work includes information on the involved mechanisms and animal models. 

1) Instead of only “Rg3”, the title and abstract should contain the word “Ginsenoside” Rg3, in order to guide readers about the article approach.

2) Please check the wording. For example in one paragraph (line 161), there are many similar phrases beginning with:

“Many studies have found that..” (…) “Several studies have confirmed that…” (…) “A recent study suggests that…” (…) “And other studies have shown that…”.

It is suggested to organize the information to avoid a "list of items" effect.

3) Line 230. “However, nanofibers also have some limitations, For example, they cannot absorb large amounts of liquid, making them unsuitable for wounds with high exudates.”

This phrase is controversial. Many authors have reported that the interconnected porosity of electrospun nanofibrous membranes is actually positive for absorbing large amounts of exudates. In turn, the swelling features will depend on the polymer composition, and it cannot be generalized for all nanofibers. Please check the following works, as examples:

10.3390/membranes11100770

10.1016/B978-0-12-820508-2.00020-9

10.1002/mabi.201300561

4) Relevant information as the dressings composition (e.g. polymer type) could be added to Table 1. Besides, it would be interesting to inform the readers about the variability of strategies for incorporating hydrophobic Rg3 into hydrophilic matrices as well as the encapsulation efficiencies, etc.

5) The first phrases in the Conclusions section can be avoided, as they were already commented on in the Introduction. Indeed, this section could be further enriched after the critical appraisal of all reported papers.

Author Response

Response to Reviewer 1 Comments

Dear Reviewer1:

Firstly, thank you very much for your comments on our manuscript. These comments are valuable for the revision and improvement of our paper, and also have important guiding significance for our research. We have taken these suggestions into account, revised the manuscript according to the recommendations and responded to related questions. The questions raised by the referees were answered and explained as follows:

Point 1: Instead of only “Rg3”, the title and abstract should contain the word “Ginsenoside” Rg3, in order to guide readers about the article approach.

Response 1: Thank you for your suggestions. According to the reviewer's opinion, We carefully considered and analyzed the title and abstract of our manuscript, and re-wrote and rearranged them.

Point 2: Please check the wording. For example in one paragraph (line 161), there are many similar phrases beginning with:

“Many studies have found that..” (…) “Several studies have confirmed that…” (…) “A recent study suggests that…” (…) “And other studies have shown that…”.

It is suggested to organize the information to avoid a "list of items" effect.

Response 2: Thank you for your suggestions. We have made a major rewrite and improvement of the manuscript.

Point 3: Line 230. “However, nanofibers also have some limitations, For example, they cannot absorb large amounts of liquid, making them unsuitable for wounds with high exudates.”

This phrase is controversial. Many authors have reported that the interconnected porosity of electrospun nanofibrous membranes is actually positive for absorbing large amounts of exudates. In turn, the swelling features will depend on the polymer composition, and it cannot be generalized for all nanofibers. Please check the following works, as examples:

10.3390/membranes11100770

10.1016/B978-0-12-820508-2.00020-9

10.1002/mabi.201300561

Response 3: Thanks for the hard work of the reviewer. We have accepted your suggestion and carefully considered and re-wrote it. ( 7.2. Nanofibers )

Point 4: Relevant information as the dressings composition (e.g. polymer type) could be added to Table 1. Besides, it would be interesting to inform the readers about the variability of strategies for incorporating hydrophobic Rg3 into hydrophilic matrices as well as the encapsulation efficiencies, etc.

Response 4: Thank you for your suggestions. We have supplemented the content of the Table 1 and article. ( 7 Rg3-loaded wound dressing )

Point 5: The first phrases in the Conclusions section can be avoided, as they were already commented on in the Introduction. Indeed, this section could be further enriched after the critical appraisal of all reported papers.

Response 5: Thank you for your comments. We have made a major rewrite and improvement of the conclusions of the manuscript.

Finally, Thank you again for the suggestions of the reviewers, which has given us great guidance. If there are still any questions, we will be patient and reply in a timely manner. Thank you for your comments on our manuscript. These comments are valuable for the revision and improvement of our paper, and also have important guiding significance for our research. We have taken these suggestions into account, revised the manuscript according to these suggestions and responded to questions relevant.

The revision have been highlighted in the manuscript, please check the attachment.

Reviewer 2 Report

After carefully reading the article titled “Polymer based wound dressings loaded with Rg3” by Jiali Yang et al (molecules-2458326), I think this work is more suitable to be a minireview than a full review article.

I also have some remarks related to:

1.   The manuscript is poorly organized, and very short.

2.   The topic is wound dressings loaded with Rg3. These contents started from the middle part of the context. Subordinate contents, such as the wound healing process, wound dressing and biological activity of Ginseng occupied too many spaces.

3.     However, the authors didn’t focus on the key issues. Actually, I can not find the significance of the polymer-based loaded with Rg3for the wound healing.

4. The advances and shortcomings of nanofibers in comparison with other ginsenoside Rg3 carriers (e.g., nanoparticles, nanosheets) should be carefully discussed.

5. An eye-catching scheme should be created to emphasize the usage of ginsenoside Rg3-loaded nanofibers for wound healing.

Minor editing of English language required

Author Response

Response to Reviewer 2 Comments

Dear Reviewer2:

Firstly, thank you very much for your comments on our manuscript. These comments are valuable for the revision and improvement of our paper, and also have important guiding significance for our research. We have taken these suggestions into account, revised the manuscript according to the recommendations and responded to related questions. The questions raised by the referees were answered and explained as follows:

Point 1: The manuscript is poorly organized, and very short.

Response 1: Thank you for your suggestions. We have made a major rewrite and improvement of the manuscript.

Point 2: The topic is wound dressings loaded with Rg3. These contents started from the middle part of the context. Subordinate contents, such as the wound healing process, wound dressing and biological activity of Ginseng occupied too many spaces.

Response 2: Thanks for the hard work of the reviewer. The topic we want to express is the application and mechanism of Ginsenoside Rg3 loaded dressings in wounds and our design ideas are as follows: We adopt a progressive writing model. First, we note that the current research on polymer dressings loaded with ginsenoside Rg3 mainly focuses on the inflammatory stage, proliferation stage and remodeling stage of wound healing course. Especially for nanofibers loaded with Rg3, most studies have discussed their mechanism in remodeling stage of wound healing, thinking that they can promote scar free wound healing. Therefore, our first part focuses on wound classification and its healing process. Secondly, the introduction of wound dressings is helpful for readers to understand the more beneficial effects of Rg3 dressings compared to traditional dressings etc. Thirdly, the introduction of the biological activity of Rg3 (not the biological activity of Ginseng) mainly focuses on the mechanisms of antioxidant and anti-inflammatory activities that be beneficial to promote wound healing, laying the groundwork for the subsequent application of Rg3 in wound healing. Meanwhile, what we want to express is that Rg3 is a highly active and low utilization drug, the emergence of polymer carriers can improve the bioavailability of Rg3. Finally, the existing types of wound dressings loaded with Rg3 were revealed and their mechanisms during wound healing were discussed.

Point 3: However, the authors didn’t focus on the key issues. Actually, I can not find the significance of the polymer-based loaded with Rg3for the wound healing.

Response 3: Thanks for the hard work of the reviewer. Our explanation is as follows: The biological activity of Rg3 shows that it has a beneficial promoting effect on wound healing, but it is a hydrophobic drug, and the polymer carrier can improve its bioavailability. Currently, various reported studies have also shown that polymer dressings loaded with Rg3 play a positive role in the inflammatory, proliferative, and remodeling stages of wound healing and we have summarized these studies and their underlying mechanisms. According to the recommendations of the reviewer, we have rewrited the article to highlight the significance of the polymer-based loaded with Rg3 for the wound healing. ( 7 Rg3-loaded wound dressing )

Point 4: The advances and shortcomings of nanofibers in comparison with other ginsenoside Rg3 carriers (e.g., nanoparticles, nanosheets) should be carefully discussed.

Response 4: Thank you for your suggestion. Our literature retrieval found that at present, polymer dressings loaded with Rg3 are mainly hydrogels and nanofibers, while Rg3-loaded nanoparticles are mainly used in the field of tumors. Please check the following works, as examples:

[1]    Qiu R, Qian F, Wang X, Li H and Wang L 2019 Targeted delivery of 20(S)-ginsenoside Rg3-based polypeptide nanoparticles to treat colon cancer Biomed Microdevices 21 18

[2]    El-Banna M A, Hendawy O M, El-Nekeety A A and Abdel-Wahhab M A 2022 Efficacy of ginsenoside Rg3 nanoparticles against Ehrlich solid tumor growth in mice Environ Sci Pollut Res 29 43814–25

[3]    Zhang S, Liu J, Ge B, Du M, Fu L, Fu Y and Yan Q 2017 Enhanced antitumor activity in A431 cells via encapsulation of 20(R)-ginsenoside Rg3 in PLGA nanoparticles Drug Development and Industrial Pharmacy 43 1734–41

The high specific surface area of nanoparticles enables them to interact with the cell surface and enter the internal environment of cells, so as to achieve high therapeutic effect. However, compared with hydrogels and nanofibers, their drug loading may be a big test. We believe that combining nanoparticles with nanofibers or hydrogels to form composite carrier to load Rg3 may be a feasible solution and we have added this section to the conclusion of the manuscript for future readers to explore and study. ( 8. Conclusions )

Point 5: An eye-catching scheme should be created to emphasize the usage of ginsenoside Rg3-loaded nanofibers for wound healing.

Response 5: Thank you for your suggestions. We have reorganized this part of the content in the manuscript. ( 7.2. Nanofibers )

Finally, Thank you again for the suggestions of the reviewers, which has given us great guidance. If there are still any questions, we will be patient and reply in a timely manner. Thank you for your comments on our manuscript. These comments are valuable for the revision and improvement of our paper, and also have important guiding significance for our research. We have taken these suggestions into account, revised the manuscript according to these suggestions and responded to questions relevant.

The revision have been highlighted in the manuscript, please check the attachment.

Round 2

Reviewer 1 Report

The comments and suggestions were duly assessed. Thank you. 
Minor typing, capitalization, wording, etc. errors remain pending.

Reviewer 2 Report

Based on the reviewers' comments, the authors made significant changes in the revised MS.  The manuscript can be accepted for publication.

Minor editing of English language required